# It’s Hard to Avoid Avoidance: Uncoupling the Evolutionary Connection between Plant Growth, Productivity and Stress “Tolerance”

**DOI:** 10.3390/ijms19113671

**Published:** 2018-11-20

**Authors:** Albino Maggio, Ray A. Bressan, Yang Zhao, Junghoon Park, Dae-Jin Yun

**Affiliations:** 1Department of Agricultural Science, University of Napoli Federico II, 80055 Portici, NA, Italy; almaggio@unina.it; 2Department of Horticulture and Landscape Architecture, Purdue University, West Lafayette, IN 47907-2010, USA; bressan@purdue.edu; 3Shanghai Center for Plant Stress Biology and CAS Center of Excellence in Molecular Plant Sciences, Chinese Academy of Sciences, Shanghai 200032, China; zhaoyang@sibs.ac.cn; 4Department of Biomedical Science and Engineering Konkuk University, Seoul 05029, Korea; p6259j@gmail.com

**Keywords:** plant growth, crop productivity, environmental stress, resource limitation, stress responses

## Abstract

In the last 100 years, agricultural developments have favoured selection for highly productive crops, a fact that has been commonly associated with loss of key traits for environmental stress tolerance. We argue here that this is not exactly the case. We reason that high yield under near optimal environments came along with *hypersensitization* of plant stress perception and consequently *early activation* of stress avoidance mechanisms, such as slow growth, which were originally needed for survival over long evolutionary time periods. Therefore, mechanisms employed by plants to cope with a stressful environment during evolution were overwhelmingly geared to avoid detrimental effects so as to ensure survival and that plant stress “tolerance” is fundamentally and evolutionarily based on “avoidance” of injury and death which may be referred to as evolutionary avoidance (EVOL-Avoidance). As a consequence, slow growth results from being exposed to stress because genes and genetic programs to adjust growth rates to external circumstances have evolved as a survival but not productivity strategy that has allowed extant plants to avoid extinction. To improve productivity under moderate stressful conditions, the evolution-oriented plant stress response circuits must be changed from a survival mode to a continued productivity mode or to *avoid* the evolutionary avoidance response, as it were. This may be referred to as Agricultural (AGRI-Avoidance). Clearly, highly productive crops have kept the slow, reduced growth response to stress that they evolved to ensure survival. Breeding programs and genetic engineering have not succeeded to genetically remove these responses because they are polygenic and redundantly programmed. From the beginning of modern plant breeding, we have not fully appreciated that our crop plants react overly-cautiously to stress conditions. They over-reduce growth to be able to survive stresses for a period of time much longer than a cropping season. If we are able to remove this polygenic redundant survival safety net we may improve yield in moderately stressful environments, yet we will face the requirement to replace it with either an emergency slow or no growth (dormancy) response to extreme stress or use resource management to rescue crops under extreme stress (or both).

## 1. Preamble

How organisms cope with environmental extremes can be traced to the distinctions between life forms explained in basic biology literature. Here the concept of life-cycles is extensively used. The two most important explanations for the universal existence of life cycles is their contributions to first, haploid/diploid switching to generate extreme genetic recombination (variation). Second, this genetic switch, during development of organisms, offers the ability to co-ordinate growth by mitosis and cell expansion with suitability of changing physical aspects of the environment for life. Even in the prokaryotic unicellular world, life cycle alterations switch between grow and no grow cycles when the environment is non-permissive to life. This resulted in the intensive selection for development of dormant spores from growing cells even without the genetic recombination advantage of meiosis. Then, after the advent of meiosis (eukaryotes) switching between spores or spore-like cells and growing cells continued. This developmental pattern was extended to multicellular organisms including higher plants that developed seeds as the dormant stage of life cycles. In addition to plants, animals, despite their mobility, also abundantly use avoidance/dormancy strategies to survive environments that are too extreme. Their developmental switches include hibernation and all the way to the extreme dormancy of tardigrades in their tun stage, which can out-survive any spore or seed (https://www.vox.com/science-and-health [1]. This use of dormancy as a developmental stage is the clear primary foundation of the success of nearly all life that faces periods of extreme environments (e.g., seasons).

We attempt to explain here how this use of dormancy in its many forms of development, not only spores and seeds but also many variations of reduced growth, form the plants stress avoidance strategies that are designed to *avoid* actual injury and death and eventually extinction. Avoiding extinction is driving the evolution of genes that control growth in threatening stress environments. Agriculturists want to use these genes to another purpose than just survival, namely productivity during stress. This use is in many ways contrary to the evolution of the functions of these genes. This is the underlying concept from which we use the term stress avoidance to explain the myriad biological forms of what we call in the literature, stress tolerance, resistance, adaptation, acclimation and even more derivative terms such as yield stability. Unfortunately, in the stress biology literature the term tolerance has been used by most plant scientists, including us, to describe plants (model systems and/or crop species) that grow exceptionally more than non-tolerant counterparts (species, cultivars and mutants) in a stress environment [2,3,4,5]. More growth under stress has been seen as “more tolerance.” However, from an evolutionary perspective, exactly the opposite is the case, less growth (avoidance of death) is more actual tolerance or more precisely, more survival [6,7,8,9]. We may therefore define *Evolutionary Avoidance* (EVOL-Avoidance) as a survival mode with the ability to avoid injury and death by various means, including dormancy. This is in contrast with *Agricultural Avoidance* (AGRI-Avoidance). This is a *productivity mode,* functional to *avoid* the EVOL-Avoidance response of which is an ancestral response, most of the time un-necessary in agricultural settings. This can be called as in our title: *Avoiding Avoidance* (Table 1). Avoiding avoidance would in nature lead to extinction. This, we suggest, is what would happen to our crops if returned to nature (without us) or, if in Agriculture, not rescued by us or rescued by hard to acquire genetics. We outline here evidence and supporting reasoning that specific genes and alleles that produce the appropriate webbed signal response in plants for abiotic stress “tolerance,” as we have been using the word in most of the literature, exist in only few species that evolved in specific environments. Strictly speaking, we have been using the word tolerance from the Agricultural and not Evolutionary perspective. We see the reason for this dominant perspective as over concentrations on Agricultural traits that we consider desirable. So, the genes involved with the interactions of plants and their environments are intrinsically interconnected with those genes that control growth, both as cell number and cell size [10]. The sensing and signalling systems involved in growth control are responsible for adjusting cell number and size to be in balance with the prevalent environmental conditions from which natural selection proceeds [11,12,13]. Agriculturists, in contrast, have selected in the opposite direction, for increased biomass and yield, with the unintended consequence that crops do not perform well under stressful conditions [14]. Wild plants exist that are less affected by the dichotomy of stress *versus* growth retardation. Even in crop species, genes exist that can be targets for modifications such that the crops will omit evolutionarily engrained growth retardation. Reduced growth retardation in response to stress, for example, can be bred for and can be engineered. However, this must be combined with management practices that prevent crops from “committing suicide” by continuing growth during progressively increasing stress exposure [15]. Although it seems possible to re-engineer and breed such plants to activate stress protection gene systems besides reduced growth, in the case of drought, the availability of water for irrigation and its prudent management will be needed and not just more genes to reach higher productivity.

## 2. *Avoidance*—The Basis of Plant Abiotic Stress Tolerance

Throughout this review, we use the term *avoidance* not as referred to in most plant stress literature but to indicate a broad genetic program (of which we know only some of its components) that sets plants in a survival mode. This involves the activation of multiple molecular and physiological responses that allow plants to avoid death (EVOL-Avoidance). This is different from the familiar ‘drought avoidance,’ a response involving adaptive traits that allow plants to maintain (relatively) higher tissue water content despite reduced water content in the soil and thus a relatively sustained growth. If “potentiated” this physiological mechanism would consent “some” growth of the plant relative to non-tolerant cultivars and/or species. This has been the underlying principle on which most genetic engineering for “stress tolerance,” as defined in the past decades, has developed [3]. Some components (traits) of the *survival mode* genetic program such as rapid life cycling with early flowering and reaching dormancy before resource limitation may turn-out useful in some agricultural contexts [16,17].

We reason here and in the following sections that plant responses to unfavourable/hostile environments from an evolutionary perspective (EVOL-Avoidance) involve activation of a *survival mode* which is fundamentally based on deactivation of the *growth mode*. The most extreme form of the survival mode is dormancy, which indeed allows plants to overcome conditions of extreme stress. Hence, potentiation of all those mechanisms that would allow some or even much growth in plants under stress (i.e., mechanisms that would strengthen tolerance according to standard terminology) is a strategy that goes in the opposite direction to the evolutionary *program* which is also polygenic and redundant because it conveys survival. Consequently, this usually generates only very marginal growth improvements under stress, often hard to see under field conditions. Stated differently, the strategy to improve plant “tolerance” (AGRO-Avoidance) from an agricultural perspective, one that we have been following experimentally in the last several decades, has been focused on forcing plants to grow against their evolutionary genetic propensity or anthropologically “against their will.” This is actually what the domestication process itself has also done. We do not advocate using the AGRO-Avoidance, EVOL-Avoidance terminology in future stress biology reports. It is intended only to help explain traditional terminology.

We will discuss here how the survival mode in crop plants is tuned to *over-protection* from unfavourable growth conditions, which is indeed often needed in natural environments to survive.’ *Over-protection* is mediated by genes that cannot risk failure to enter the survival mode as rapidly as needed in nature. Rapid entry into the survival mode is not usually needed in most agricultural environments where stresses are actually much less in intensity and do not usually persist for time periods long enough to threaten survival. Unravelling the genetic link between reduced growth and stress will open new research avenues to reduce un-necessary over-protection in plants in standard agricultural settings and to potentiate safety mechanisms that are deployed by plants and by us, only under extreme stress that actually threatens plant survival, a condition quite rare in most agricultural contexts [12,18,19,20].

Therefore, this may be a good time to re-think thoughts that have shaped our views of the standard view of plant abiotic stress “tolerance” concept and interpretations of our expanding knowledge. Indeed, much information has been accumulated, much of it interesting and often correct in detail but real genetic based understanding has not yet come from the more than sixty (physiology) or forty (molecular), or twenty (genetics or genomics) years of work on the topic [21,22,23].

The “physiology and molecular years” have been spent mostly on a few specific crops and easily cultivated models. With respect to a focus on what is called in standard terminology plant stress ‘tolerance’ or its opposite, sensitivity, the recent “genetics and genomics revolution” has been centred almost entirely on the model Arabidopsis, which as has been pointed out is not a stress “tolerant” species [4,24]. Nevertheless, genetics and genomics concepts and several technological innovations applied to Arabidopsis have opened new vistas and paradigms, essentially by studying genes that result in an even more stress-sensitive Arabidopsis plant when altered [25,26]. In turn, it is becoming easier to apply what we learned with Arabidopsis to other plants [20]. Turning to species adapted to extreme environments can teach us much about what will allow implementation of successful agronomical practices over a broad range of challenging environments [27,28,29,30,31,32,33,34]. We reason that we might better look for superior characters (e.g., those that uncouple slow growth and stress response) in species that are evolutionarily adapted to thrive sometimes even with high productivity in extreme habitats and challenging climates [35,36,37,38,39,40,41]. It is there that we will likely be able to find what has eluded detection up to now by using the advanced technologies. Recent work in rice has shown that desensitization of plant response to ABA, through inhibition of specific components of the ABA/PYL receptors system, may enhance plant growth while reducing the negative effects of stress responses [19]. What we will find among these evolutionarily adapted species will be mechanisms that are largely made up of ubiquitous genes that have been used in novel ways to avoid, or work around, problems posed by the environment [41,42]. We suggest that plants lack genuine (in the conventional sense) tolerance-conferring genes and proteins but that success in coping with disastrous events comes almost universally from the presence of genes and proteins mediating mechanisms that induce the survival mode (our term—EVOL-Avoidance). These are molecular/physiological traits that allow plants to survive and go to the next generation. These same traits are not necessarily key to improve crop productivity under suboptimal growth conditions. These mechanisms evolved in plants necessitated by the daily struggle to recognize deviations between homeostatic internal optima and continuously changing external environmental conditions that threatened the very survival of a species over long time periods. The true, accurate avoidance concept can be grasped by considering that even a so-called “tolerance” protein must manage not to undergo an environmentally-induced conformation change, denaturation, or inhibitory interaction originating in the environment [43]. Likewise, considering the true nature (what has evolved) of “stress tolerance” (our EVOL-Avoidance), the thought has emerged that the same genes conferring protection to high stress in naturally “tolerant,” avoiding species that grow slow are also present in sensitive species that grow fast [39,42]. The emergence of plant life on land relied on a set of genes present from the beginning, whereas evolution re-used these genes or modified versions (alleles) to produce functions (phenotypes) not needed by fresh water adapted plants (glycophytes). These genes returned to creating species capable of halophytism again. These retro-type species began once again their pre-land occupation functions occupying niches, such as estuaries and salt marshes. A ramification of this view is that, first, all species could be or become either very fast or slow growing species under stress. Second, we have in the past most likely not dealt with really important genes or versions of genes (alleles) since we have only partially succeeded in isolating those components that are introduced and/or duly modify in important crop species could improve their ability to grow fast under stress (ability to avoid avoidance, in the context of agricultural settings, see Table 1). This view, we think, is not in conflict with data on neo-functionalization of genes, for example, starting with alternatively spliced or otherwise processed and increasingly sequence-divergent and so forth isoforms, that can escape from adaptive conflict by further directional selection after duplication [39,41,42,44,45,46,47,48].

The standard tolerance—avoidance terminology has long ago been introduced [49,50,51,52]. Again, we do not advocate supplanting all of it with new terms like *avoidance* and *avoiding avoidance*. However, we emphasize that the mechanisms employed by plants to naturally cope with a stressful environment resulting from evolution are overwhelmingly geared to *avoid* detrimental effects that can lead to injury and death. The new genetic, genomic and bioinformatics approaches and data collected from them have been consistent with the view of avoidance as a unifying concept.

## 3. Survival and Avoidance of Injury and Death Are Negatively Correlated with Growth, Biomass and Productivity

The first noticeable casualty of a changing environment is the growth rate. It is often described as being the most sensitive response to stress [53]. Stress may be viewed as a condition that does not permit optimal growth, measured in comparison to the growth rate of the species or genotype in optimal conditions. Here, it must be clear that “experimental” growth in a controlled, optimal environment, phytotron, growth chamber, or culture plate should not be considered practically comparable with growth in natural settings; many typically stress-induced genes are highly expressed in a seemingly stress-free environment under field conditions [54,55]. In a changing environment, growth modifications occur, either quantitatively (biomass/cell divisions), or qualitatively (switch in developmental phase) or both [10,56,57]. Clearly, an altered environment may lead to resource limitation, which then necessarily limits growth but this is also based on a genetic hard-wired program aimed at avoiding a non-sustainable situation. Recent studies showed that the activity of Target-of-Rapamycin (TOR) kinase, the central regulator of energy and biomass production, is inhibited by osmotic stress and abscisic acid (ABA) treatment through SnRK2 mediated phosphorylation of Raptor [58,59], indicating the presence of a pro-active growth control system activated by environmental stresses. Similarly, Zhao et al., [57] have shown that the loss of CYCLIN-DEPENDENT KINASE C2 function increases cell division during leaf development and enhances plant growth under drought stress. From an evolutionary perspective, however, survival is what is established in a changing environment by natural genetic responses.

Plant growth shows distinct phases of cell divisions, extremely fast or imperceptibly slow cell expansion and many degrees and stages of maturation and reproductive development. The genetic expression guiding those events are pre-programmed, mediated by hormones and fine-tuned in response to the environment by receptors and many layers of signal control networks. It is environmental fine tuning that concerns us because the degree of the effect of external conditions on growth is not hard-wired but follows a malleable program with hard-wired limits [59]. In principle, except for situations of extreme resource limitation that can lead to death, there is no functional reason for a plant to initiate slow growth (under moderate and/or transitory stress) other than that it has the genes that direct such a response as an evolutionary survival mode. Arguably, many responses to stress have been interpreted as strictly adaptive, whereas we suggest that most actions are general alarm reactions, or responses aimed at preventing the inevitable resource limitation. However, plant hypersensitivity which activates these alarm reactions before reaching resource limits prevents the achievement of full growth potential under moderately/mild stress. This occurs quite commonly throughout a growth season [3,60,61,62]. If carefully examined, alarm reactions can be revealed as attempts to enter the survival mode (grow slow/low productivity/avoid death) more rapidly. The most observed and obvious alarm is stomatal closing. The most extreme alarm response is senescence and dormancy [57]. Plant biologists have used genes involved in the alarm reactions to gain “tolerance” (AGRO-Avoidance) by overexpressing them. This approach has led only to incremental, or marginal, advances toward our ultimate goal. This and the complexity of stress responses, has also resulted in many different, incomparable data appearing in the literature, including measures of plant survival [63], an important trait under stress but not very helpful in most agricultural contexts. Indeed, increased alarm reactions do usually lead to better survival and subsequent growth after stress dissipates, just as has occurred in evolution. Such genetic variants are good at avoiding death or presenting an appearance (e.g., less injury) that we associate with better “tolerance” or growth (AGRO-Avoidance) [12,57]. The resurrection plant, which enters a vegetative dormancy before reproductions, represents an extreme alarm reaction that is the result of natural selection in an environment where survival required it. In specific agriculture settings it also could be important as is escape [64,65].

A fundamental concept re-emerges here. As dictated by the environment, stress-responsive genes have evolved not for biomass production but as required for survival. Commonly overlooked is that survival genes act in response to conditions that are usually rare in agricultural contexts (production environments). Their functions developed over a long time scale, a scale that by far exceeds the ten millennia of plant culture. We argue that the defect or absence of a gene that is needed only once in many years for the survival of a species—these may include genes controlling dormancy under extreme drought events [66] will lead as easily to extinction as a gene needed daily (e.g., genes involved in stomatal regulation [56]). This concept has a historical foundation. Dobzhansky (1964) has famously stated that “nothing makes sense in biology except in the light of evolution” [67]. This has inspired our effort to re-examine what we mean by “tolerance.” Earlier, Darwin envisioned biological diversity and plasticity in terms of natural selection and on the basis of heritable diversity upon which natural selection would work, although some rethinking is taking place here in light of a new look at epigenetics [68]. It has taken Mendel’s breakthrough and more than 100 years of experimentation to know that there are still “unknown unknowns”. The most obvious unknowns are the various members of the RNA world and epigenetic information [68] that add until recently unanticipated control layers [69,70].

We think it reasonable to modify the often used axiom of ‘survival of the fittest’ and change it into ‘what we see today is what has survived.’ However, what has survived can only be compared to a crude paleontological record, revealing important logical gaps in our view of evolution: (1) Fossils are our only way of knowing what has not survived. Records of [CO_2_] trapped in ice cores, for example, provide valuable but insufficient data points. (2) We cannot reconstruct in any detail the environments in which the non-survivors failed. We barely know essential details of today’s environments.

Certainly, experiments to test hypotheses about extinctions are not possible. Even in fields of science where direct experimentation is possible, results may lead to hypotheses that give rise to erroneous conclusions. There are two important concepts we would like to put forward: first, survival is the ultimate measure of evolutionary success, not productivity. Agricultural ‘success’ cannot depend on survival alone but must be concerned with productivity. Seeds represent exquisite survival mechanisms; they represent the ultimate wager. It is however not the number of seeds that counts; rather it is the seeds that produce new plants that are the guarantors of survival. Second then, if increased productivity carries even the slightest disadvantage for survival over a long time period, it will be selected against. Therefore, limits of a productivity phenotype cannot be predicted in a strictly evolutionary context. A corollary of this fact is that productivity ranges in any environment for presently existing species are not predictable as well, because the ranges in productivity are genetically irrelevant for survival. Therefore, they must be empirically used only to reveal the underlying genetic structure and plasticity of the species in question.

The crop domestication process provides valid illustrations for these statements. The original wild populations of our domesticated species had nowhere near the productivity of later domesticated cultivars even under optimal conditions. A remarkably small number of genetic changes was necessary to bring about the very large increases in productivity in non-stressful conditions. These few gene alterations coupled with adjustments in agricultural practices designed by the domesticators allowed early farmers to maximize the genetic benefits [71,72,73]. In rice, in fact, the green revolution was brought about by a single mutation in a gene later identified as encoding a GA-synthesis gene [74,75,76,77]. When combined with better fertilization practices, this gene brought about the substantial increase in productivity in the “green revolution” [78]. Our existing crops already depend on human management to survive and any newly developed crops will most likely be even more dependent on human intervention in their growth.

Environmental control of the productivity range or limits of a phenotype are ultimately determined by resource availability. Resource limitations that are potentially associated with environmental stress includes those that have typically been noted, such as reduced amounts of [CO_2_] when stomata are closed, or water supply when stomata are open, or supply of any essential nutrients. However, we may also add process constraints among the resource limitations. For example, these can be limits to altered membrane transport to accelerate ion homeostasis in saline or water deficit conditions and stress-dependent altered hormone homeostasis, or metabolite disequilibrium (C/N) conditions. Following this perspective, many genes and pathways undoubtedly contributed to the phenotype of abiotic stress avoidance.

## 4. Plant Growth versus Survival and Productivity

There is a strong correlation between success, viewed as survival, in extreme environments and natural slow growth, even constitutive slow growth in the absence of stress. We note that this correlation is not absolute and that exceptions have been discussed [12,20,79,80,81,82]. We state that a focus on this correlative dichotomy has distorted our views of the true nature of the linkage between stress and productivity. Importantly, the correlation strongly implies that many species have not evolved satisfactory stress EVOL-avoidance mechanisms that allow fast growth in moderately and still persist in more extreme, stressful environments. Corollary to this consideration is that slow growth and success under extreme stressful conditions, although linked in complex fashion, may conceivably be separated genetically [10,12,56].

We emphasize that this very tight correlation has caused previous considerations to often confuse stress EVOL-avoidance with resource limitation. A good example of the perplexingly strong linkage between compromised growth and stress without resource limitation is provided by describing responses to NaCl relative to water deficit [79,83]. In saline environments water is almost always plentiful but thermodynamically difficult to retrieve and thus requires a genetic response to adjust metabolism to the thermodynamic imbalance, whereas water in a dry environment is limited and may approach zero, thus becoming a limited resource, yet slow growth is tightly linked to both stresses. Similar constraints apply to all micro- and macro-nutrients and to the extreme ranges of temperature or light that restrict growth or induce altered growth and development. Within extreme ranges, however, the altered slower growth phenotype must remain linked to a genetic response program that assures survival even at the expense of growth before the extremes that cause resource limitation and ultimately death, are reached. Exposure to conditions that exceed life chemistry limits, such as taking more water to cool a larger plant, more ions to sustain growth and more energy allocation to fight biotic challenges are avoided to a large degree by small plants. Thus, large plants are less able to utilize micro-niches or to generate sufficient protection from environmental extremes. Productivity is not the issue under such conditions. The correlation between stress and reduced growth emerged from physiological or natural selection paradigms, not from the insights provided by experiments and artificial genetics, an idea that has been difficult to amend let alone discard.

At the risk of overstating, in our crops, as in most species, stress reduces growth such that biomass accumulation (overall productivity) and yields and reproductive productivity are almost always compromised. We argue again that the negative correlation between growth in a stressful environment and biomass is mostly based on an EVOL-avoidance strategy in that, importantly, “anticipates” encountering non-survivable environmental conditions [10]. The relationship of reduced growth to any experimentally defined version of AGRI-avoidance is presently confusing relative to EVOL-avoidance because these AGRI-avoidance versions almost always neglect the survival advantage or experimental sensitivity (Table 1). Equalizing these coincidental but conflicting views of the relationship of growth to the conventional term tolerance (AGRI-avoidance) without considering survival reactions to stress environments (EVOL-avoidance) may have hindered the path to understanding the true importance of different genetic mechanisms. Consequently, we are compelled to again state that slow growth results from being exposed to stress because genes to adjust growth rate to external circumstances have evolved as a survival (EVOL-avoidance), not productivity (AGRI-avoidance), strategy (and not a strategy of altered growth only) that has allowed extant plants to necessarily avoid extinction (EVOL-avoidance). Consistent with this statement, recent research has clearly revealed in unprecedented detail, the reciprocal regulation of ABA and TOR signalling that controls several processes in balancing plant growth and stress responses [58,59].

## 5. The Energy Limitation Myth

An analysis calculating the efficiency of photosynthesis for converting solar energy into biomass has been conducted and the differences between achievable maxima and biomass productivity have been discussed [84,85]. Values for C3 and C4 plants emerging from this work were below those that constitute a theoretical maximum. However, the values, based on well-researched bottlenecks and inevitable energy losses, are still higher than what amounts to the average conversion efficiency shown by our major crops. The authors attribute this difference to “unfavourable physical environments”—stresses. The calculations allow for another conclusion. The evolution of land plants over approximately 450 million years has, as far as can be gathered, not resulted in favouring increased conversion efficiency to make use of what is available. Differently stated, it seems that plants did not need to evolve toward improving energy use efficiency in using a resource that is almost always virtually unlimited. This convincingly indicates that energy is not a limiting factor in most ecosystems. That plants do not make use of what is available and often make do with much less, is certainly stress-dependent but we see this as a relationship associated with fewer and smaller cells, or no cells at all, that plants generate by the EVOL-avoidance or survival mode response over long periods of stress. This strategy, again, is geared at avoiding extinction and not to avoid excess energy expenditure. The smallness concept, resulting in lower productivity, has been discussed for a hundred years or more [4,86,87]. Others have also deduced that the “energy” problem most likely also reflects the long-term—on an evolutionary scale—avoidance of extinction by resource limitation [88]. We consider this critical strategy as almost never based on energy resource limitations. A striking example is provided by true shade plants that will not grow faster in higher light—a condition that appears to represent the same slow growth phenomenon (EVOL-avoidance) seen in many other abiotic stress conditions, that are not initially resource limiting. Such conditions include water deficit, salinity, temperature extremes, declining nutrient availability and even high light stress that can induce biochemical damage. These stresses generate limits only over time but the decline in plant growth is often much more instantaneous. Accordingly, less energy consumption almost certainly reflects a survival strategy acting over a period of time much longer than agricultural settings when a larger energy allocation—as bigger plants consume more resources—eventually reaches a limited state. Thus any genetic program that continues to guide increased biomass will require even larger resource allocations not to combat stress environments but to avoid in the long run constraints that will likely lead to extinction (see halophytes—discussed in [4,35]. Again, plants have evolved to respond early and be prepared for challenges by the environment and even often have constitutive genetic programs to meet these potential environmental assaults. Plants embarking on a route that results in relatively increased total biomass, fruit or seed, persevered less well under stress conditions and became extinct not only because of impending resource limitations but also because they became bigger targets for pathogens and predators and supported a much larger expansion of plant species populations even allowing more competition between species. This is also true within a species population that becomes a larger threat to species survival. This is adequately supported by the common observation of within species allelopathic inhibitions in extremely dry environments. Thus, we see the conservation of energy in extant species serving an entirely different purpose (involving long-term survival) that is not related to the notion that energy is often limiting or even close to limiting in agricultural time frames in almost any stressful environment.

An important aspect related to the energy consideration is harvestable resource allocation or, rather, the problem of harvest index and yield stability under stress [53,89,90,91]. In fact, maize breeders unknowingly selected for *avoidance of stress avoidance*, AGRI-avoidance, by selecting against increased competition caused by higher planting densities, through analysis of yield stability, defined as the overall highest yield under a range of different environments at different planting densities [89]. If actual resource limits can be circumvented by applying appropriate agronomical practices, some species can grow under severe stress conditions at rates that are equal or close to those of a typical crop species under non-stress conditions [24,80,92]. In other words, extremophile species can exhibit naturally, astoundingly high yield stability [93,94,95]. This fact provides the major justification for paying attention to the growth rates with and without stress of particular plant species that are close relatives of the Arabidopsis model and of extremophile versions of crop species [33,34,81,96].

## 6. A New Conceptual Framework

We consider now the plant abiotic stress archetype by incorporating a genetic-genomics viewpoint that has only more recently become possible. Unlike in the past, we have now available genome and transcriptome sequences and collections of tagged or otherwise generated genetically altered lines that reveal by their phenotypes the underlying function of the altered genes. In addition, the gene silencing and editing technologies are further revolutionizing our abilities [97]. Genes that respond to abiotic stresses, based on microarray and RNA sequencing technologies coupled with statistical analyses [6,7,9,29,98,99,100,101,102,103,104,105,106,107,108,109,110], show significant overlap between different stresses and their nature has helped to identify pathways and networks that are affected by different stresses [6,108,109,110]. The many examples of past attempts at genetic engineering included modifications of osmotic adjustment (betaine accumulation), ion transport (CAX, HKT1, SOS1, NHX), [41,42,111], membrane potential (ATP synthases), radical oxygen scavenging and redox control (SODs, catalases). Also, morphological traits [112], hormone homeostasis (ABA, IAA) and signalling (receptors, protein kinases/phosphatases), [12,18,82,113,114,115,116,117,118,119], or transcription factors involved in the activation of various pathways have been exploited [52,120,121]. In many cases some avoidance of stress avoidance improvement (AGRI-avoidance) has been observed, which often could be repeated and quantified [52]. However, with a few exceptions [3] the engineered lines have not stood up to the test in the field [122]. In addition, presumably protective effects disappeared over several generations owing to our unawareness of the importance of epigenetic control circuits that respond to environmental forces. Potentially, practical avoiding of stress avoidance (AGRI-avoidance) has been defeated also by agriculturally negative phenotypes such as undesirable altered flowering times [123] linked to AGRI-Avoidance through complexity of signal networks. The next generation of stress AGRI-avoidance engineering, we surmise, will utilize the greatly enhanced knowledge and the gene resources available [124]. For example, a condition-specific promoter has been already, long ago, engineered to control induction of a gene leading to altered hormone homeostasis [125] that has also been shown to mediate responses to limited water [12,126] Other entities, high in the hierarchy of genes and pathways that may be strengthened are being explored, with transcription factor and receptor genes high on the list [124,127]. Engineering approaches are rapidly being surpassed by gene editing technologies that have addressed the common redundancy of hormone mediated signalling [19] and the very complex regulation of plant growth versus responses to environmental stresses, which may involve additional/unidentified components of osmotic stress signalling [128]. These technologies may even present allele conversion capabilities because gene editing can produce multiple SNPs (Single Nucleotide Polymorphisms). The most successful strategies in our opinion will include the modification of pathways that govern meristem activities, including flowering, by disconnecting or countermanding the connection between stress sensing and slow growth initiation. The NFYB transcription factor from maize provides a fascinating example of future approaches [127]. Upon over-expression, it suppresses (causes *avoidance of avoidance*, that is, AGRI-avoidance) the slow growth phenotype that is the typical response to moderate water deficits. The plants no longer ‘anticipate’ and respond by EVOL-avoidance to an increasingly stressful environment that could lead to death over a longer time period. A short duration and moderate non-resource limiting drought period would result in these engineered plants having increased biomass and yield (AGRI-avoidance) as reported and more recently confirmed under field conditions [127]. We do not know whether these plants would experience some survival problems in the absence of agricultural management (e.g., due to impaired stomatal closure at advanced stress conditions). Similar results have been obtained via overexpression of bacterial RNA chaperones in corn, a strategy that has led to the development of the only commercial corn line for improved grain yield under water-limited conditions [129]. In agricultural environments such stress/slow growth (EVOL-avoidance) decoupled plants will require the addition of precise resource management or deployment only under strict, stable climates to avoid catastrophic crop failures (see Beachy R. [130]). The increasingly variable weather patterns associated with global climate change is exacerbating such strategies.

Four fundamental physiological processes have emerged from decades of abiotic stress research that should help guide future abiotic stress genetic engineering approaches and attempts. These processes and importantly the signal networks that control them are especially important for performance in drought, osmotic and salinity stress conditions since they compose the major responses that allow plants to avoid serious injury and ultimate death. These include (a) osmotic and ionic adjustments [131,132], (b) radical oxygen induced injury containment [23], (c) other aspects of metabolic homeostasis [21,133,134] and (d) growth control [59]. These mechanisms are controlled by hormone-mediated signalling and therefore this approach must be considered on different levels of “intervention urgency,” with the higher-order hormone control circuits and signal networks of superior importance.

In this category, two major aspects require attention. First, we need more knowledge about how to disengage damage response aspects of signal networks from the growth arrest and growth maintenance aspects of growth networks. Second, the opposing or combining effects of stress hormone alterations on both EVOL-Avoidance (slow growth) and on developmental thresholds, for example an inappropriate flowering time or pollen silk interval and control of the root/shoot growth ratio in connection to stress [135], must be understood more fully at the genetic level.

By moving the community’s focus onto the concept AGRI-avoidance as earlier defined rather than premature alarms (EVOL-avoidance) (Table 1) we hope to stimulate discussion for which we have provided here arguments and a justification for viewing plants as having evolved genes and phenotypes for EVOL-avoidance stress survival. Again, we do not advocate changing existing terminology, only how we think about this terminology. The following are some of the areas of consideration that we advocate to be discussed in efforts to refocus.

(1) On an organismal level we need to distinguish the roles in stress of the mechanisms that operate in different tissues and cells and during developmental phase changes that define windows of intervention.

(2) Rather than using blinders that exclude whatever is outside our narrow focus, we need to recognize that the sum of effectors that respond to the environment should be visualized as a web or field of vectors which often ‘pull’ the phenotype into different directions, perhaps in a manner where multiple vectorial components do not act independently and may antagonize or potentiate each other. We point to the example of a spider’s web, where the spider unerringly moves to the place the web’s contact with prey that may be far from the spider’s position and even out of its view. Instead of a “linear thread” that leads directly to the spider, the spider produces a “web” a net or orb shaped structure that can capture prey from many directions and positions. This contact signal initiates a highly specific wave (dependence on the point of contact) that reaches the spider and is perceived and processed (this is poorly understood) to reveal the exact position of contact whereupon the spider moves directly to this position. It appears that the spider has evolved signal network processing perhaps like computer learning or even a crude version of the human brain. We should also keep in mind that both of these processing networks have limitations. We need to know much more about how webbed pathways (interconnected signalling pathways) function and are controlled perhaps by comparisons to neural networks.

(3) Such a field of vectors acts not only through ABA or ethylene or auxin, or salicylate or jasmonate but certainly through suppressors and activators mediated by hormone combination. A complete hormonal profiling and signalling molecules in various genetic backgrounds is of highest priority because hormone networks manifest the major phenotypes controlling stress avoidance and survival [136,137]. Hormone receptors usually have the first contact with an exogenous signal. They exert the most dramatic effects through the webs of developmental and external genetic response programs. Even though some receptors and many intermediates such as kinases, phosphatases and so forth have been discovered, many signal components remain elusive. Especially understanding the roles of interconnections (cross-talking) perhaps will well reveal a mechanism for re-wiring as in computer learning and human memory—which involve likely epigenetic mechanisms.

(4) Genes have names that have been given by researchers but the genes may have and many do, have functions that can become obscured by focusing on the labels we contrived. Annotations should never be confused with functions, even with assignments based on phenotypes because they are still most likely incomplete. Genes can have more than one function and they may have different functions depending on the cellular, temporal and conditional context in which they operate [41,138]. A lucid example of the latter has been provided by a study of amyotrophic lateral sclerosis in a mouse model. The mice, over-expressing a SOD1 gene perplexingly both potentiated the disease and were found to respond positively to an inhibitor of SOD1 activity. It was later found in a more rigorous study that these results were explained because the SOD1 enzyme possessed another previously unknown activity that generated other more toxic ROS [139]. This and several other cases [140] have shown that the DNA sequence does not always reveal its true function.

(5) Stress response networks operate on several levels of control, including many that are largely unknown, especially the epigenetic components that have only recently and gradually come into focus [141,142,143]. Stress adaptation must be also viewed as another epigenetic state such as operates during ontogeny where signal networks may be formed in different patterns [68,69]. The manifestation of many EVOL-avoidance mechanisms, through classical and epigenetic means, have generated the present biodiversity of species that escaped extinction and established ecosystems in a wide range of climates and conditions.

(6) Irrespective of this apparent complexity, we have been encouraged by the surprisingly large phenotypic effects exerted by individual genes emerging from Arabidopsis screens. Yet, it must be remembered that the degree of effect of a single locus on a complex trait depends dramatically on its genetic background (other genes) that can be equated with the evolutionary history of each species. Yield, certainly a complex trait, has been significantly increased by very few domestication genes that were necessary to convert a wild species into a crop that now depends on our care [71,144]. The trait, flowering time, also illustrates very well the ability of a single locus (or of very few loci) in the appropriate background to have major effects on complex traits. Such phenomena have been recognized as the “Mendelization” of quantitative traits to which one might add that all traits have a quantitative component [4,145,146].

(7) Nevertheless, these examples underscore the real possibility that such critical loci affecting positively and negatively EVOL-Avoidance exist. As we have promulgated here, their criticality to survival has almost certainly made them redundant. The formidable late generation sequence technologies and advanced correlative algorithms have allowed them to be exposed to us now. Needed then, are also more inclusive databases (and the curation of misleading annotations therein). Because the information content is getting quite large, we need better statistical methods to sort and categorize and continuously alter and improve the mathematical tools and their applications. The iPlant concept (http://iplantcollaborative.org/) supported by the National Science Foundation was a very early undertaking in the direction of an integrated database of all plant knowledge, which will require additional consideration of this process. Additional examples include international initiatives on data publishing such as FAIR a Findable, Accessible, Interoperable and Reusable data sharing platform [147,148]; the PLAZA platform (https://bioinformatics.psb.ugent.be/plaza) a plant-oriented online resource for comparative, evolutionary and functional genomics [149]; software to perform genome-wide association studies (GWAS) in a variety of species [150]. The correlation of phenotypic metadata with the genome structure of a large number of Arabidopsis accessions and mutants, for example, has been in the past a high priority [151]. Also, we need scrutiny of the observational quality provided for several non-crop model species and crops because metadata of older experiments and annotations are often lacking or incorrect. We need integration of transcriptome dynamics, as well as protein amount, turnover, activity control and structure. We also need more information about metabolic pathway interactions and finally the epigenetic status of each stress adaptation state of a crop species. Only large databases can present such information in a cogent manner and preferably in the form of interactive network models which are still ongoing endeavours.

Virtual experiments can be generators of ideas, provide clues, predictions and hypotheses that can initiate (well-planned) wet lab experiments, eventually replacing assumptions with reality-tested rules. This may even drive new algorithms to more accurate predictions for future experimental efforts.

## 7. Genomics, Bioinformatics and Integration

Research areas tend to reach stagnation and conceptual deadlock and become in need of rejuvenation. A major problem it seems is the narrow view that results from focusing on a specific, usually easily defensible, hypothesis promising incremental progress in a well-researched field. In fact, funding agencies have insisted on caution and have rewarded timidity. Still focusing on advancing fortunately, Arabidopsis has produced formidable sophistication in molecular and genetic tools for this species that allow us to study stress in an integrative fashion to trace the input factors to their genetic roots and apply them to crop species.

As a note of caution, over-emphasis on technical abilities can diminish conceptual progress. Abiotic stress is not to be simply understood by, for example, measuring ABA and the expression of all genes responding to ABA. Improving biomass or yield under stress does not come from simply knowing all components that are involved in the stress response. As we have pointed out, most observed responses are symptoms and indicators of deviations from homeostatic equilibrium with any particular environment. The importance of such responses should be carefully pre-examined (e.g., by edited knock outs) before deciding our level of commitment to them.

That Arabidopsis has become the primary stress model is based on superior understanding of the plant’s development and the tools that have been developed for analyses. As a stress tolerance (EVOL-avoidance or AGRO-avoidance) model (either or both grows faster under stress or survives higher stress) this plant is arguably inappropriate or at least insufficient. What we have done is make the plant genetically more sensitive (grows slower dies sooner) to a stress and then record the genetic entities that are associated with such engineered sensitivity (Table 1) [152]. In some cases of such approaches, more so in development and in the gene-for-gene background of biotic stress responses connected to innate immunity, we have generated finely tuned compromised genetic lines [153]. However, this approach also can go only a limited distance before the interconnections of these responses eventually become obviously limiting to our progress (as previously observed in gene for gene disease resistance that cannot be transferred to other species and cold tolerance that is difficult to genetically identify in other species [133,153]).

Needed are different models. Models that are true extremophiles and genotypes with either stress EVOL-avoidance or fast growth (AGRI—Avoidance) and especially different species that display either mechanisms that have been provided by evolution will finally yield real world solutions tailored to very specific ecologies and climates. For example, fast growing marine species (e.g., sugar kelp) can provide good sources of some important loci and alleles [40]. These models should also, for now, if possible exhibit some features that make Arabidopsis so easily studied and manipulated. Such models are found among close relatives of Arabidopsis. We have termed them Arabidopsis Relatives Model Systems, ARMS [31,81]. Yet we need to go further still, for example, to wild relatives of rice [154], or cotton [155]. In fact, we believe that such models can be found in each crop family and that we can expect family-specific stress tolerance (EVOL-avoidance) mechanisms that will paraphrase mechanisms that unite plants. The genetic programs supporting stress avoidance of avoidance (AGRI-avoidance) apparent in uncommon wild species can now be reasonably approached with whole-genome forward and reverse genetics, genome-wide association mapping combined with the identification of essential or important loci by whole genome sequencing and re-sequencing [36,37]. This will, we expect, provide targets for more precise gene editing especially targets that are redundant such as PYL receptors [19]. In essence, we consider that one organism’s intolerable stress can be another one’s normal condition and that comparative studies of extremophiles are an excellent way to finding the ways that disengage productivity from the slow growth survival mode [19,30,32,35,96].

## 8. Thinking around the Dogmas—Searching for Answers where the Problems Are

Crop yields have increased tremendously in the last 100 years, yet the yield potential of crops is still at least some three times higher than what the best lines and best agricultural practices can accomplish today [89,90,156,157]. Irrespectively, modern lines have steadily increased yields even in less than optimal environments. This “breeding” coupled with appropriate agricultural management largely has been responsible for the increases. An obvious conclusion based on past progress is that our approaches to stresses, especially abiotic stresses, including water deficit and ion imbalance, have let us make only incremental improvements whereas fundamental progress is what farmers, industry and politicians expect and what is desperately needed to feed a growing more affluent global population [130,158,159].

To achieve increased food security calls for unprecedented cooperation between molecular geneticists, breeders, agronomists, geologists, hydrologists, economists and scientists of many other disciplines [160]. Political leaders will have to recognize that food shortages will dwarf the numerous experienced devastations caused by earthquakes, volcanoes, tsunamis, floods, hurricanes and such and will occur before we will be hit by an asteroid and before global climate change exacerbates temperatures, polar ice cap melting and sea levels rise enough to intensify these many problems. When lack of fresh water becomes more prevalent and when hunger and starvation strikes, all other problems, goals and aspirations become irrelevant [161].

We have argued here that the evolutionary process converged on growth control as the mechanism most reliable for the survival of a plant species. This is diametrically opposite to the demands we are placing on our crops. Genetic studies that have been directed towards understanding AGRO-avoidance we again emphasize, point towards the lack of actual tolerance mechanisms. How could a plant become tolerant to the absence of water? Plants must avoid the absence of water or remain in some dormant state (e.g., seed, spore, etc.). There is that old adage—plants cannot grow without water [162]. It should be stated again and clearly understood that the AGRO-Avoidance “tolerance” mechanisms that are usually cited, stomatal closure, enhanced root to shoot growth ratios, epidermis wax layers or trichome growth or altered orientation of the leaves, in our view, are not mechanisms that in themselves make plants AGRO-avoidance “tolerant” [57] and grow more under stress. It is clear that evolution favoured survival (EVOL-Avoidance) and almost never AGRO-Avoidance “tolerance” (Table 1), which by necessity in evolution, resulted in genetic programs that direct plants to initiate EVOL “avoidance.” This response almost universally and importantly reduces growth through activating growth repressing or deactivating growth inducing developmental programs. In evolution, growth (or metabolic) control quickly becomes the key process in any actual or potential resources limited state. The difficulty in finding a mechanism to achieve desired growth control (faster) can be compared to a strategy where plants are “hedging their bets,” to be small, grow slowly and to conserve available resources, such as water, that would become limiting earlier in a large plant. This view has inevitably led us to understanding that stressed plants do not initially grow slowly and produce less because of actual resource deficiencies or energy limitations [4,10]. Exposing the fallacies surrounding energy and growth limitations we think, can reveal possibilities to more effectively control development that directs energy and growth to the specific parts of the plant that are useful that is, root growth in water deficits or harvest index in all stresses that initiate reduced growth (harvest) [68]. Moreover, stress-adapted species, which can be found in many plant families that support this approach as well [28,81,163,164]. In fact, in the case of saline adapted species, some species are known [165,166,167,168,169], especially those natural to estuaries that have constitutively uncoupled slow growth and stress response. These can grow almost equally fast with saline stress or without it—see fast growing species at river estuaries [40] and halophytic perennial species already used as forage grass [170]. It is here that we may find the most important coupling factors. This will be made much easier with new technologies like massive sequences, ortholog searches and CRISPR approaches to test functions [19].

In addition, a reverse strategy is re-emerging to make species that are naturally EVOL-Avoidance tolerant more productive (AGRI-Avoidance tolerant). Examples include Salicornia and Opuntia [171,172] while new ones like Quinoa are exploiting newer genomics technologies [173] offering renewed optimism.

Various editorials and essays have discussed the discrepancy between the urgency to conserve water, the necessity to generate better, more stress-adapted crops and the perceived failure by plant molecular biology and gene engineering to deliver [130]. The *more-crop-per-drop* catchphrase may have inspired but it also may have confused the underlying concepts and diverted from the optimal strategies about how to circumvent inevitable water shortages (Table 1). To produce more food with less water, or less suitable water, will depend on adequately supported collaborations between the many disciplines that must become involved. This has consequences for agencies, because the funding of new concepts has been almost invariably declined by programs where thinking out of the box is most often discouraged.

Exceptions that charted new ground have been few in the past. These are projects such as the SALK insertion tagging project that has fundamentally altered and advanced our approach to genetics by providing the genomic information to explore the myriads of phenotypes from countless mutant screens [174]. The TAIR project has collated information from many different approaches initially for Arabidopsis but now for other species as well [175,176]. Of similar foresight and impact was the 1001 genome project [151]. The project continues to provide the genome sequences and phenotypes from ecotypic, evolutionary imprinted variation in the genus Arabidopsis. Importantly, it has revealed both genomic and phenotypic information derived from natural variation and will allow us to expand our knowledge of the genetic details of the solutions that survived evolutionary scrutiny. We will obtain data for many genotype/phenotype comparisons that have been missing, causing a proliferation of hypotheses but perhaps still not having the benefit of sufficient data from diverse and important species. Much more creative effort is needed to provide useful algorithms that can mine these data, especially to identify orthologs and paralogs of genes that have functions known in model species but not in important crops.

Hopefully, the evolution oriented stress response genetic programs will be exchanged with an enhanced productivity under stress genetic program. If we are to use genetic manipulation of our crops to produce or yield more in environments that are even moderately stressful, we must finally recognize that plant responses to stress almost always are not, at least at initially, because of any resource limitation. We think that we can succeed in improving yield under moderately stressful conditions *by removing the premature alarms* that trigger growth reduction and other unproductive responses that are certainly polygenic and redundantly programmed in plants that have assured their survival in evolution. But, we will need to keep a stronger, delayed developmental response like meristem (bud) dormancy that occurs in summer dormancy [12], or use resource management to rescue them, or both [12,57] under extreme stress conditions.

## Figures and Tables

**Table 1 ijms-19-03671-t001:** Relationships of tolerance nomenclature with growth, stress and survival.

**In Nature Setting**		
**Grow Phenotype**	**Terminology**	**Survival Phenotype**
Grow slow	Tolerant	Avoidance of death—lives
Grow fast	Sensitive	Avoids avoidance of death—dies
Grow fast	Tolerant	Avoidance of death—lives
	The phenotype in Nature that we seek among species	
**In Agriculture/Experimental Setting**		
**Grow Phenotype**	**Terminology**	**Survival Phenotype**
Grow slow	Sensitive	Avoids avoidance of death—dies
Grow fast	Tolerant	Avoidance of death—lives
	The phenotype in our experiments and screens that we seek and desire for Agriculture use	
Grow slow	Sensitive	Avoids avoidance of death—dies
	The phenotype that we actually search for in almost all of our experimental screens	

Conflation of the wording describing tolerance as growing fast and sensitivity as growing slow in agriculture and/or experimental stress settings, while describing fast and slow growth in an evolution context in the opposite way has resulted in confusion. However, the genetic basis of stress tolerance in both Agricultural and Experimental settings and in Evolution has remained the same. That is, genes from evolution evoke tolerance by growing slow and thus avoidance of death. But! to be stress tolerant as it is called in Agricultural/Experimental settings we must find the genetic bases of fast growth under stress conditions but also avoid the eventual death caused by fast growth when stress becomes too extreme. Such plants that grow fast under moderate stress and still avoid death when stress becomes too extreme are rare but they do exist. A brief conceptual synthesis to reframe our search for key mechanisms that allow plant growth under stress follows: (1) Our known genes are almost always from experimental screens for mutants that grow slower and die faster (loss of function screen). (2) From this we conclude that the wildtype allele of this locus is required for full tolerance. (3) We usually conclude that such required genes may convey more tolerance when overexpressed ectopically. This is true but only incremental gains have been achieved. (4) Screening also for mutations that affect stress-induced marker gene expression has been very successful. Most of these mutants do display a stress phenotype also. (5) These required low and altered expression loci have provided information to build models of the signal network that controls plant responses to stress and the genes that the network controls. (6) The genes or alleles of genes that control our desired phenotypes are being used by plant species that display the search target phenotype (grows fast, called tolerant and does not die at high stress). (7) We need opposite screens, ones for more Agriculture/Experimental tolerance (grow faster/dies slower). These are gain of function screens. (8) Single genes can rarely convey a gain of function; therefore, single mutations in single genes are very unlikely to result in faster growth and dying slower (gain of functions). (9) Using molecular genetic tools with such search target natural species with the correct phenotype has not been plausible before. Now our technologies do allow us to carry out gene searches of these species (that have been previously very difficult to use experimentally) by several new and old approaches.

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
