# Peer review of "It’s Hard to Avoid Avoidance: Uncoupling the Evolutionary Connection between Plant Growth, Productivity and Stress “Tolerance”"

_ijms, 2018, doi:10.3390/ijms19113671_

Round 1
Reviewer 1 Report
Manuscript is very well written, and I don’t seem any reason for not accepting it. It will attract large audience of IJMS.
Few minor corrections:
L178 fourty to forty.
L187 paradymes or paradigms.
L308 paleontologyical to paleontological.
L345 examples, these to examples, there
L538 L539 Recheck line.
L573 an SOD1 to a SOD1.
L567 to L568 make line compact.
L598 effecting to affecting.
L599 exist to exists
L631 A of Arabidopsis should be capital.
L639 pre-examined or re-examined?
L651 genes is repetitive.
L716 approaches to approach.
Author Response
Dear Editor,
Attached please find the revised manuscript based on the indications of the anonymous reviewers, which we found helpful for improving the quality of our review. Below please find a point by point reply to the reviewers’ comments. In brief:
1) All changes/edits proposed by the two reviewers have been considered and included in the attached revised draft.
2) The reference list has been formatted based on the journal guidelines.
3) In addition, we edited/added a few sentences for improving clarity and some additional references for completeness. All these changes are also traceable in the track-changes document.
4) In this revised version of the manuscript, we also included a cleaner/slightly modified version of Table 1.
Best regards,
DJY
RESPONSE TO REVIEWERS – Reviewer Comment (RC); Our Response (OR)
Reviewer1:
Comments and Suggestions for Authors
Manuscript is very well written, and I don’t see any reason for not accepting it. It will attract large audience of IJMS.
Few minor corrections:
RC: L178 fourty to forty.
OR: Done - Line 229 in the Track-Changes version.
RC: L187 paradymes or paradigms.
OR: Done - Line 237 in the Track-Changes version.
RC: L308 paleontologyical to paleontological.
OR: Done - Line 370 in the Track-Changes version.
RC: L345 examples, these to examples, there
OR: We could not find this.
RC: L538 L539 Recheck line.
OR: The sentence has been edited - Line 632-633 in the Track-Changes version.
RC: L573 an SOD1 to a SOD1.
OR: Done - Line 667 in the Track-Changes version.
RC: L567 to L568 make line compact.
OR: Done - Line 662-663 in the Track-Changes version.
RC: L598 effecting to affecting.
OR: Done - Line 694 in the Track-Changes version.
RC: L599 exist to exists
OR: Done - Line 695 in the Track-Changes version.
RC: L631 A of Arabidopsis should be capital.
OR: Done - Line 727 in the Track-Changes version.
RC: L639 pre-examined or re-examined?
OR: OK “pre-examined”, i.e. examined in advance before further committing in following a specific strategy - Line 739 in the Track-Changes version.
RC: L651 genes is repetitive.
OR: Done – “gene for gene” has been replaced by “single gene”- - Line 751 in the Track-Changes version
RC: L716 approaches to approach.
OR: We believe “approaches” is ok since it refers to many ways we can use CRISPR technology - Line 835 in the Track-Changes version
Reviewer 2 Report
It’s Hard to Avoid Avoidance: Uncoupling the 3 Evolutionary Connection Between Plant Growth, Productivity and Stress ”Tolerance”
Albino Maggio et.al, 2018
Comments for author: It is an excellent review to read. As a plant stress scientist, it was always in my mind how seed dormancy (Hibernation) could play role in organisms evolutionary or adaptation during stressful events. The author simply nails all details about adaptation, stress tolerance or stress avoidance. Stress caused death, stress caused stimulation of survival for the future and make fit for future generation. Recently, plant stress scientist discovered genes that could memorize their bad time (stressed) to ready for future upcoming stress events. Overall the manuscripts are well written. It would be great if the author could make a diagram of stress elicitors stress tolerance or avoidance (EVOL-Avoidance/ AGRI-avoidance), and a table of different crop species in terms of adaptation and evolutionary connections review. Minor edits are also important to improve the quality of this manuscripts.
L 105 – L120: All this information should move down to L-129 and also some linkage sentences should add to make it comprehensive for a reader, as right now all this information elucidate without adequate elaboration.
L-125 and L-126: Agricultural/Experimental should change to Agricultural and Experimental; Agricultural or Experimental
L-142: (EVOL‐ Avoidance) Space problem
L-173: Zhao et al., 2016b, ”b” needs to delete
L-178: (genetics/genomics) should be (genetics or genomics)
L-256: CYCLIN‐DEPENDENT KINASE C;2 (CDKC;2) need to remove ;
EVOL‐avoidance or EVOL‐Avoidance: the author should write either one of the formats
AGRI-avoidance or AGRI-Avoidance: the author should write either one of the formats
L-460: RNA Seq: should write RNA sequencing
L-487: Zhao et al., 2016b should remove b
Some of the citations need to make homogenous format, i. e. citation 3, the title is all in cap letter, citation 7, Scientific reports should be Scientific Reports, thus cap letter and small letter problem present all the reference section. Also, the author should follow one of the formats either write abbreviated journal name or full name, sometimes the journal name abbreviated in others full name formatted. I would suggest the author should use one of the reference citation software and make each reference same format.
Author Response
Dear Editor,
Attached please find the revised manuscript based on the indications of the anonymous reviewers, which we found helpful for improving the quality of our review. Below please find a point by point reply to the reviewers’ comments. In brief:
1) All changes/edits proposed by the two reviewers have been considered and included in the attached revised draft.
2) The reference list has been formatted based on the journal guidelines.
3) In addition, we edited/added a few sentences for improving clarity and some additional references for completeness. All these changes are also traceable in the track-changes document.
4) In this revised version of the manuscript, we also included a cleaner/slightly modified version of Table 1.
5) With respect to the suggestion of reviewer 2 “It would be great if the author could make a diagram of stress elicitors stress tolerance or avoidance (EVOL-Avoidance/ AGRI-avoidance), and a table of different crop species in terms of adaptation and evolutionary connections”, we believe this is premature. We would like to emphasize that the purpose of this review was to open a new perspective in approaching the study of stress tolerance in plants. Due to overlapping and connections of stress responses it would be difficult and approximate, at this stage, to make a “classification” of EVOL-Avoidance vs. AGRI-avoidance elicitors. We clearly stated that we used this terminology only instrumentally to re-interpret old terminology and introduce new concepts – as we wrote at line 157-159 “We do not advocate using the AGRO-Avoidance, EVOL-Avoidance terminology in future stress biology reports. It is intended only to help explain traditional terminology”.
Best regards,
DJY
RESPONSE TO REVIEWERS – Reviewer Comment (RC); Our Response (OR)
Reviewer2:
Comments and Suggestions for Authors
It’s Hard to Avoid Avoidance: Uncoupling the 3 Evolutionary Connection Between Plant Growth, Productivity and Stress ”Tolerance”
Albino Maggio et.al, 2018
Comments for author: It is an excellent review to read. As a plant stress scientist, it was always in my mind how seed dormancy (Hibernation) could play role in organisms evolutionary or adaptation during stressful events. The author simply nails all details about adaptation, stress tolerance or stress avoidance. Stress caused death, stress caused stimulation of survival for the future and make fit for future generation. Recently, plant stress scientist discovered genes that could memorize their bad time (stressed) to ready for future upcoming stress events. Overall the manuscripts are well written. It would be great if the author could make a diagram of stress elicitors stress tolerance or avoidance (EVOL-Avoidance/ AGRI-avoidance), and a table of different crop species in terms of adaptation and evolutionary connections review. Minor edits are also important to improve the quality of this manuscripts.
RC: L 105 – L120: All this information should move down to L-129 and also some linkage sentences should add to make it comprehensive for a reader, as right now all this information elucidate without adequate elaboration.
OR: Done – L 121-129 in the Track-Changes version.
RC: L-125 and L-126: Agricultural/Experimental should change to Agricultural and Experimental; Agricultural or Experimental
OR: Done - Line 124 and 126 in the Track-Changes version.
RC: L-142: (EVOL‐ Avoidance) Space problem
OR: Done - Line 154 in the Track-Changes version.
RC: L-173: Zhao et al., 2016b, ”b” needs to delete
OR: We left “b” since we have 2 references “Zhao et al., 2016”. These are different Authors (Zhao C and Zhao Y) therefore we would follow the editorial office if we have to leave it in the text as Zhao et al.2016a and Zhao et al., 2016b or we have to report the initials Zhao C et al., 2016 and Zhao Y et al., 2016. Line 187 in the Track-Changes version.
RC: L-178: (genetics/genomics) should be (genetics or genomics)
OR: Done - Line 229 in the Track-Changes version.
RC: L-256: CYCLIN‐DEPENDENT KINASE C;2 (CDKC;2) need to remove ;
OR: Done - Line 311 in the Track-Changes version.
RC:EVOL‐avoidance or EVOL‐Avoidance: the author should write either one of the formats
OR: Done – Edited as Avoidance throughout the text.
RC: AGRI-avoidance or AGRI-Avoidance: the author should write either one of the formats
OR: Done – Edited as Avoidance throughout the text.
RC: L-460: RNA Seq: should write RNA sequencing
OR: Done - Line 539 in the Track-Changes version.
RC: L-487: Zhao et al., 2016b should remove b
OR: See explanation above – the ref is now at line 574.
RC: Some of the citations need to make homogenous format, i. e. citation 3, the title is all in cap letter, citation 7, Scientific reports should be Scientific Reports, thus cap letter and small letter problem present all the reference section. Also, the author should follow one of the formats either write abbreviated journal name or full name, sometimes the journal name abbreviated in others full name formatted. I would suggest the author should use one of the reference citation software and make each reference same format.
OR: The entire reference list has been formatted according to the journal guidelines.